# Social Capital in Relation to Mental Health—The Voices of Adolescents in Sweden

**DOI:** 10.3390/ijerph20136223

**Published:** 2023-06-25

**Authors:** Mikael G. Ahlborg, Jens M. Nygren, Petra Svedberg

**Affiliations:** Department of Health and Welfare, Halmstad University, 301 18 Halmstad, Sweden

**Keywords:** adolescents, content analysis, mental health, networks, qualitative, relationships, social capital

## Abstract

The social environment that adolescents interact in has undoubtedly changed over the past decades. The latent constructs of social capital that have been described in theory may be universal, but it is necessary to reveal sociocultural specific pathways and manifestation in order to validly operationalize social capital for adolescents. There is a call for qualitative data to enhance our understanding of social capital for adolescents today and the specific sociocultural context they live in. The aim of this study was to explore social capital from the perspective of adolescents in relation to mental health. Twenty-three semi-structured interviews were conducted in a school setting with a sample of adolescents aged 11 and 15 years. Qualitative content analysis was applied, and analysis remained on a manifest level. From having adolescents describe their social relations and networks in relation to mental health, three main categories were formed: accessing a safe space, with sub-categories of trusting enough to share, having someone close to you, and being part of an inclusive and honest environment; feeling connected to others, with sub-categories of hanging out and having things in common; and maintaining control, with sub-categories of deciding for yourself, dealing with change, and having social skills. Having access to a safe space is vital for adolescents’ mental health, by providing resources such as mutual trust, honesty, and unconditional access. Feeling connected to others is important in close relationships and reveals the glue that holds networks together, but also links to sociability in a wider sense. Predictability in adolescents’ social relationships and networks, influenced by internal and external factors, may be a resource of increasing importance in todays’ society and an interesting subject for intervention and future research on social capital and adolescent mental health.

## 1. Introduction

Despite a growing general awareness of the negative trends for mental health among adolescents, the prevalence of mental health problems has continued to rise over the past two decades, and somewhere between 10 and 20% of adolescents globally experience mental health problems [1]. A review of cross-national studies reveals the trend to be especially worrying in the northern part of Europe [2]. The use of interaction-based interventions in the school and community setting has shown promising effectiveness in reducing mental health problems, at least in the short-term [3]. One explanation for this could be the strengthening of social capital at the individual and the community level since many of the protective factors for adolescent’s mental health represent social dimensions that have been described as ‘social capital’. Social capital has been defined as the sum of resources that individuals access through their social networks [4]. Systematic reviews of quantitative research on the social capital of adolescents speaks in favor of a positive relationship with health in general [5], but a dark side of social capital that holds social contagion has also been shown [6]. Earlier research has concluded social capital in the family, school, peers, and community to be associated with various mental health outcomes [5], pointing out family social capital as a consistent protective factor [7,8,9]. There are, however, limitations in existing quantitative research that relates to design and consequently what conclusions can be drawn from these studies. The limitations relate primarily to the cross-sectional nature of data, which limits the possibility to prove causality and disentangle the meaning of a transforming social network in adolescence. Moreover, the breadth of the concept of social capital aggravates assessment that would allow for a thorough mapping of resources in different networks and its impact on mental health, or vice versa. Heterogeneity in definition, conceptualization, and consequently the measurement of social capital in relation to adolescent mental health is also evident [10], and it is common that studies do not conceptualize social capital for adolescents as distinct from that of adults [11]. This leaves us with knowledge that has been argued to be over-simplistic [11], and that is not easily transferable into practice, whether the goal is to strengthen social capital or prevent mental health problems among adolescents.

The social context that surrounds young people today is fundamentally different from what existed throughout the late 20th century when theories and conceptualizations of social capital, mainly for adults, emerged. These changes relate mainly to the advancements of digital communication that have erased the boundaries of spatial distance and time of physical interaction, while simultaneously increasing adolescents’ exposure to potentially harmful content and cyber-bullying [12]. This is an unsupervised arena for adolescents to interact in that has merged with their physical environment, which plays an important role in their identity exploration and social development [12]. It has been suggested that the definitions and conceptualizations of social capital used to guide research on adolescents may be influenced by adult perspectives and outdated conditions [13], and in light of the lack of qualitative studies within this field, that is most likely also the case today. Adolescents, to a great extent, have not been considered active agents in forming their own social capital [14]. The original theories and previous conceptualizations undoubtedly hold value for our understanding of social capital, but it has been suggested that quantitative research on social capital needs to build on the qualitative findings that exists to date [15]. There is, thus, potential in the use of qualitative methods to elucidate any sociocultural specific pathways or manifestations within different social networks and contexts [16]. During the last decade, adolescents have mainly been the target of quantitative longitudinal and population-based studies as informants, while involvement in qualitative research is scant. There are a few examples of exploration of how adolescents describe the resources that are social capital. For example, a qualitative study that explored social capital in relation to wellbeing from adolescents’ point of view [17] revealed a particular sociocultural context, signified by strong bonding ties and a lack of bridging and linking social capital. A recent systematic review showed that qualitative elements are largely missing in the development of social capital instruments [10], and only a handful of the identified instruments that were developed and validated for adolescent samples involved adolescents in the development phase through qualitative work.

To increase the understanding of adolescents today and their perspectives, researchers are calling on qualitative studies that let adolescents have their say on matters that clearly concern them [18]. The potential value of increasing adolescent involvement in research that aims to enhance the understanding of the relationship between social capital and mental health are a higher relevance, effectiveness, quality, meaningfulness, and impact of findings [19]. In doing so, research arguably needs to change in focus from conducting research “on” or “about” a target group, to “with” them. A higher level of active involvement of the target group can help advance social capital theory [16], perhaps especially for social capital among adolescents as something distinct from that of adults. In that way, it allows for a conceptualization and operationalization of social capital that is more sensitive and culturally appropriate with higher validity and reliability [19]. Such involvement is especially important during the initial stages of the research process before critical decision making closes the opportunity for higher-level participation [20]. As a response to this, the present study has an inductive qualitative design that aims to explore social capital from the perspective of adolescents in relation to mental health.

## 2. Materials and Methods

### 2.1. Study Design 

For this study, a qualitative design with an inductive approach was used. Data were collected through individual interviews between May of 2019 and February of 2020. The present study applied a conventional content analysis, described as an appropriate design when the research literature on a phenomenon is limited [21]. The method is defined as a “subjective interpretation of the content of text data through systematic classification process of coding and identifying themes or patterns” (p. 1278) [21]. By following the systematic procedure of content analysis, researchers are able to create an understanding of the studied phenomenon and develop new or extend pre-existing knowledge of human experiences. This procedure adopts an inductive approach where open-ended questions are used when data are gathered through interviews and predetermined categories, or labels are avoided in the analysis. The study follows the Consolidated Criteria for Reporting Qualitative Research 32-item checklist ensuring its trustworthiness [22].

### 2.2. Setting

In Sweden, the school system comprises 10 years of compulsory education and 3 years of voluntary high school and optional higher education. Pupils begin first grade at age seven, meaning they most commonly turn 16 the year that they finish ninth grade. The national setting represents a modern welfare state with a largely secularized population and socialized education and healthcare. School absenteeism (>5% of schedule) has been estimated at 2% among 5th graders and about 11% among 9th graders in 2020 [23]. For this study, pupils in the 5th (11–12 years) and 9th (15–16 years) grade at three primary schools in the south-western part of Sweden were included. These ages for inclusion were chosen because early adolescence has been described as a period of potential for setting a trajectory for health and health-related behaviors into adulthood [24]. Additionally, research shows that half of all mental health conditions onset before late adolescence [25]. A purposive school sampling technique was applied to include schools that represent both smaller and larger cities and to increase the probability of a diverse sample of pupils in terms of socioeconomic status and living environment. However, due to the spread of COVID-19 in March of 2020, school sampling was compromised and one school that was based in a larger city was unable to allow recruitment of participants. Therefore, included schools were located in one smaller municipality, with a total population of about 26,000 inhabitants of which about 14 per cent were born abroad, which is comparable to the national average of 16 per cent. Two of the schools (range of 300–370 pupils) were in the largest city in the municipality, while one school of about 550 pupils was in a smaller city. 

### 2.3. Recruitment and Participants

The director of the school district was informed about the aim and procedure of the study and given the chance to ask any questions before signing a consent form for recruitment within that school district. Principals in each of the three schools were then contacted and informed about the aim and procedure of the study. Then, teachers were consulted by the principals about the current workload and were given the chance to ask questions related to the study. After agreeing on a time and place, one researcher (MA) from the research group visited the school classes in question and gave oral information to each class about the study and answered any questions before handing out written information and consent forms. Pupils under the age of 15 were given additional information and consent forms for guardians. All written information was in Swedish. Pupils were given a period of one to two weeks to hand in their written consent to their teachers who handed them over to the researcher (MA) upon their next visit. After all consent forms were collected, the researcher (MA), in consultation with teachers, decided on an appropriate day and time during school hours for conducting interviews that would have as little impact as possible on schoolwork. 

All pupils in the 5th and 9th grade at each school who were present during the day that the researcher visited were given oral and written information about the study, while teachers collected written information to hand out to absent pupils. In total, 30 pupils, out of about 300 eligible, gave their informed written consent, 10 from 5th grade (11–12 years old) that provided additional written consent from guardians and 20 from 9th grade (all 15 years old). Due to the spread of COVID-19 in March of 2020, seven pupils could not be interviewed since the schools did not allow the researchers further visits. Therefore, a total number of 23 interviews were conducted, six boys and four girls in 5th grade and six boys and seven girls in 9th grade. All participants had good to excellent skills in the Swedish language. Participants represented different levels of socioeconomic status (parental employment), family constellations (divorced parents, bonus family members), religious views, and different living conditions (rural, urban, apartment, house). The sample also included pupils with parents born outside of the European Union and pupils with learning disorders in need of assistant teachers in the classroom. 

### 2.4. Data Collection

The interviews were conducted in small study rooms at each school, made available by teachers with a guarantee of not being disturbed. The rooms were situated so that windows did not face any areas where pupils went during recess, to ensure privacy and to create a more relaxed atmosphere. The interviews were semi-structured and an interview guide with open-ended questions was used to give participants the chance to talk freely about what social relations and networks were important in relation to their mental health, what made those relationships important, and how they were distinguished from other relationships. Interviews lasted between 17 and 58 min (median = 26). Before interviews started, all participants were asked if they had any questions about the study and were again asked about their willingness to participate to ensure voluntariness. The interviews were audio recorded digitally and transcribed verbatim. Interviews started by asking adolescents to talk a little bit about themselves. Next, participants were asked what they considered to be the underlying meaning of the response “I am fine”, when asked the question “How are you?”. The interview then proceeded with questions such as “what persons do you consider to be important for you to feel fine?”, “what is it in your relationship that makes you feel that way?”, and “can you give examples of what you do together with that person?”. Examples of follow-up questions were “could you tell me a bit more about that?” and “can you give any examples of that?”. 

Four interviews were conducted in May of 2019, originally intended to be solely pilot interviews. Minor changes were made to the interview guide after pilot interviews, mainly by including an open question asking the participants to consider what people they considered important, accompanied by minor changes of phrasing of other questions. However, since data gathering was cut short due to the outbreak of COVID-19, these four interviews were reviewed once more and considered to be rich in content relating to the aim of the study, and thus suitable for inclusion in the analysis. The proceeding 19 interviews were conducted between November of 2019 and February of 2020. 

### 2.5. Ethical Approval and Informed Consent

The study protocol that elaborated on method, recruitment of participants, interview procedure, information on confidentiality, and data management was approved by the Swedish Ethical Review Authority (Dnr: 2019-00068 & 2020-00741). Participation was voluntary and informed consent was sought from adolescents, as well as from their guardians if adolescents were below the age of 15. The procedure followed the principles of the Declaration of Helsinki [26]. All respondent names in the results section are fictional.

### 2.6. Analysis

The analysis followed the detailed process put forward by Erlingsson and Brysiewicz [27]. First, the transcribed interviews were read several times to get a sense of the whole. Then, the text was divided into meaning units. Meaning units were discussed in the research group regarding their relevance to the aim of the study. The third step was to condensate meaning units. One author (MA) conducted the condensation, but condensed meaning units were reviewed amongst all authors and discussed to ensure the meaning was not lost in the process. Fourth, codes were formulated by labeling all meaning units with a few words to describe the content. Initially, coding was mainly carried out by one author (MA) and discussed simultaneously with one other author (PS). After the first round, the coding process included all authors since discussions led to several revisions with the aim of keeping codes on a more manifest level. When consensus was reached about coding, the fifth step was to group codes into sub-categories followed by main categories. Throughout steps four and five, codes, sub-categories, and categories were continuously compared with original transcripts to ensure the meaning was not lost. Categories and sub-categories were discussed and refined several times until consensus was reached among authors. The analysis was conducted over several months to ensure thoroughness and refinement of interpretations. All authors agreed that analysis would stay on a manifest level (Table 1).

## 3. Results

When adolescents described their social relations and networks in relation to their mental health, what emerged as important was (a) having access to a safe space, (b) feeling connected to others, and (c) being able to maintain control in their social interactions. When these needs were met, adolescents felt happy and that life in general was going well, while also feeling confident to manage negative events that could arise in relationships or networks.

### 3.1. Accessing a Safe Space

Having access to a safe space emerged from adolescents’ portrayals of their closest relationships that contained aspects that they felt were vital for their mental health, namely trust, honesty, and unconditional access. Through their descriptions, it became obvious that adolescents evaluated their relationships continuously, put effort into strong relations, and that the safe space was not static. Differences in experiences related to gender and age were seen in this category. Sub-categories consisted of “Trusting enough to share”, “Having someone close to you”, and “Being part of an inclusive and honest environment”.

#### 3.1.1. Trusting Enough to Share 

Adolescents described how trusting others was a prerequisite for sharing feelings and problems or seeking support in relations and networks, but also that trust was built through dialogue and sharing. Thus, sharing in this context was not about everyday topics or small talk, but about sensitive matters that were shared very selectively with others. To be able to confide in friends, parents or significant others, adolescents described certain expectations that needed to be met. For example, they spoke of the need to feel confident that the receiver(s) could keep a secret and not tell others. Relationships with parents, siblings and friends were described to meet this expectation but also extended family, such as a cousin, aunt, or grandparent. While boys mainly described parents as their first source of support, girls, and especially the older girls, expressed that they could not talk to their parents about everything. They felt that parents could be trusted and offer support, but that they would not understand certain matters, or that they would give poor advice. Older adolescents described sharing a deep bond with siblings to a greater extent than younger adolescents. Turning to siblings for advice meant a great deal to them. Most adolescents mentioned the importance of a best friend or a small group of close friends, but it was mostly the older girls that saw peers as their first source of support. Adolescents exercised a sort of ranking of trust in relations by stating a first choice for sharing, then a second and so on. This ranking of safe space was dynamic depending on, for example, the issue, who was involved, and what feelings were triggered. As “Jenny”, 9th grade, described it:

Jenny: *How I feel has probably a lot to do with my friends … I can really talk to them about …. not feeling well … when it comes to everything … while if it’s my parents then I can say like this … it’s tough now … it’s stressful in school and so … but I perhaps can’t talk about everything with them.*

In relation to who could be trusted and why, some girls explicitly stated that they would never talk to boys about sensitive matters. This was not necessarily due to a lack of trust but instead because boys in general were incapable of talking about their feelings and would therefore not understand or be able to offer support. The same reasoning was used by some girls when talking about seeking support from parents and how they preferred going to their mother over their father.

Relationships with teachers or school counsellors could also entail trustworthiness. One adolescent described how the bond to the school counsellor had become very strong after her family had gone through some rough times. She expressed how, because the counsellor knew what she was going through, she felt more at ease during school hours. Other adolescents, however, described how they would not share personal matters with teachers, because there was no trust between them. Some of the older adolescents explained that the trust they felt in certain teachers began forming when the teacher hade shared personal matters with them, and they felt that this was a sign of trust from the teacher since this information could very well be used against the teacher. For some adolescents who attended sports practice frequently, the relationship with their coach had become strong and they could discuss family matters or talk about things that troubled them. 

Trusting enough to share was described in different forms. Mostly described as feeling safe to talk about anything with another person or group face-to-face, but there were other examples. The trust could also extend beyond sharing personally into an understanding that between family members or within a circle of close friends, sensitive matters could be passed on without the presence of all involved as long as what had been shared initially did not leave the circle. As “Charlotte”, 9th grade, explained regarding trust between her closest friends: 

MA: *So, if you say something to one of these people then this person won’t tell it to someone else?*

Charlotte: *Then it’s like … I have two friends in another class and one friend in my class, it’s those three who I talk to the most … and so if I talk about something to one of them in the other class … it depends on what it is …but then the other can find out but then I’ve been on the way to telling her as well so it’s ……*

MA: *But it doesn’t go outside the group?*

Charlotte: *No, not if they’ve asked me whether it’s okay …*

When adolescents described what made them trust others enough to share, some said the reason was because the other person shared sensitive things with them. Mutual sharing was therefore another expectation that needed to be met to sustain trust. Teachers that shared personal issues with adolescents were considered more trustworthy and therefore easier to confide in. Some adolescents emphasized how having a long history together meant that the other person knew much about you, which facilitated mutual trust. However, having a long history together was not enough in itself for trusting enough to share. There were examples given of why adolescents had lost trust in certain relations or networks. Adolescents drew attention to how bad experiences in friend networks, e.g., feeling abandoned or friends not keeping a secret, could lead to long-term distrust that instead resulted in a stronger sense of safe space within the family. Moreover, in some networks adolescents would not trust members with sensitive issues despite being a group of close friends. “Peter”, in 9th grade, explains how he felt he could not talk about feelings with a group of friends:

Peter: *Eh … well then …we often tell jokes with a few other friends about everything being so serious so it can be easy to … if you start talking about feelings and so on … that they just take it as a joke and … because we usually joke about such things …*

#### 3.1.2. Having Someone Close to You

Adolescents described relationships or networks that were characterized by an unconditional access or sense of safety related to a belief that the other person(s) was there for them if they needed. This did not imply that the relationship was flawless, exempt from disagreements or quarrels, but instead that they were able to count on someone when it really mattered. In broader terms, having someone close to you, unconditionally, was described as an essential part of maintaining mental health. Adolescents’ feeling of having someone close, whether it was friends, family, or even a pet or a belief in God, prevented them from feeling lonely, which was a genuine fear expressed by many. In friend networks, this meant knowing friends would not abandon you, that they cared about you no matter what, and that you had support whenever you needed. When in school, adolescents who had established a trusting connection with the school counsellor expressed a feeling of always being welcomed there, which made them calm. While some older adolescents described this in abstract and hypothetical ways, the younger adolescents preferred giving concrete examples. For them, being close meant knowing friends would comfort you if you injured yourself or friends having your back if you came into a fight or an argument with someone. As “Michael”, in 5th grade, described it:

Michael: *They care about me … if … if I hurt myself, they’ll come straightaway … and then I know they’re good friends and so I know that they care and so they help me and such ….*

Within the family, being asked about your day on a daily basis gave adolescents a sense of relief knowing that there was a recurrent and unforced opportunity to talk to someone if needed, instead of having to seek support actively. Commonly this opportunity was presented during family meals or when parents helped with homework. This made adolescents feel that parents were always there for them. “Jim”, 5th grade, states:

Jim: *They [parents] are kind to me, they’ll never push me away sort of and … yes, they’re most often there when I need them*

MA: *How do you notice that they’re there? Is there anything special that they do?*

Jim: *Oh, they just care about me without me having to say to them like … yes*

#### 3.1.3. Being Part of an Inclusive and Honest Environment

When defining aspects of what constituted a safe space, adolescents described an atmosphere that characterized certain relationships and networks. Examples of feelings in that environment were feeling welcomed, safe, and loved and not feeling left out. Complete honesty was brought up when speaking about spending time with close friends, even when it meant somebody’s feelings might get hurt. The environment also included not feeling judged and feeling you could be yourself. These descriptions were primarily attributed to family, extended family, and friend networks. Being able to joke about anything was also synonymous with an inclusive and honest atmosphere. “John”, in 9th grade, described the atmosphere between him and his assigned teacher assistant and other teachers:

John: *yes, a guy who is a personal assistant we usually banter when our football teams are not playing well or so … and those who I talk most with or so are my mentors and we have built up so that we can joke and stuff like that one can talk about this and that …*

Adolescents contributed to creating this environment themselves through their actions and behavior, but other circumstances also had a great impact. In particular, girls described how the atmosphere in a group was dependent of its size and individual participants. The atmosphere could change simply by one friend entering or leaving a group when friends were hanging out. One girl suggested that in order to discuss sensitive matters, a smaller number of friends was preferable. If more friends joined, it could discourage her from bringing up personal issues. The same argument was brought up by other adolescents in relation to online chat groups. Both written and spoken conversation in groups of predetermined members seemed to provide the same opportunities as meeting in real life. Some adolescents characterized the atmosphere in these online groups as supportive and positive, while others described them in a more general sense as a place to talk about anything. 

Examples were given of networks where adolescents felt excluded or where dishonesty characterized the environment. One boy who previously had experienced bullying both in his football team and in school avoided groups due to difficulties trusting others, even old friends. Instead, he had formed a close bond with players in a new football team. Others talked about the downsides of living in a small community, with rumors spreading fast between peers, parents, and networks. This was a source of uncertainty that made adolescents more appreciative of their close relationships that were characterized by honesty and were less prone to socializing outside of their immediate peer group. Adolescents also expressed how a calm atmosphere in their neighborhood made them feel safe. Friendly neighbors and an absence of disturbance or violence contributed to this sense of safety, which led to a belief that people in general were friendly minded. “Michael”, in 5th grade, stated: 

Michael: *There are some [neighbors] that I haven’t met but I think that all of them are kind …*

MA: *Why do you think they’re kind?*

Michael: *Because … if you go round and say hello … they usually sort of ask how your day has been like ... so that’s it …*

### 3.2. Feeling Connected to Others

Feeling connected to others revolved around connectedness in close relations and networks but also in a broader context, aspects that helped shape the identity of adolescents and give them a sense of belonging and purpose. Sub-categories were “Hanging out” and “Having things in common”.

#### 3.2.1. Hanging Out

Adolescents described how hanging out increased their sense of belonging and connectedness. Doing things together with family members, friends, and others was considered positive in many ways. Apart from not having to feel alone, hanging out meant enjoying things together, sharing laughs, being spontaneous, remembering the past, and making plans for the future. It seemed as if hanging out increased adolescents’ sense of identity.

Hanging out was described in various ways; in school, it could mean anything from playing games together with the entire class to spending time alone with your best friend. Riding the bus home from school was considered important because it meant everybody was talking to each other and having fun together. Outside of school, it included activities such as going to the gym, sports practice, taking a walk, or simply sitting next to each other with smart phones showing each other funny video clips. Some adolescents described how hanging out with friends entailed connectedness and how it was essential for their well-being. Whether it was with friends, family members, or close relatives, simply being together with others could mean a great deal to them. “Filip”, in the 5th grade, who had struggled in school in the past, described what spending time with his grandmother meant to him:

MA: *Okay, what’s it like being with her [grandmother]?*

Filip: *It’s really good … sometimes it’s … her and my little sister and myself and then we might go shopping like … but it’s really good also …*

MA: *Okay … what’s good with that?*

Filip: *I don’t know … it just feels good to sort of get away sometimes … just to be with my grandmother so … and my little sister …*

Hanging out also included a non-physical dimension. When playing computer games, for example, the possibility of having simultaneous online voice conversations exclusively with friends added a social dimension that by some participants equated gaming alone at home to sitting next to multiple friends. While these online conversations were described as meaningful, some adolescents pointed out that the physical dimension was lacking when hanging out online. The online setting gave an opportunity to continue social interaction after school without the physical presence of others that seemed to be taken for granted. Adolescents reflected on how the opportunity to hang out online meant the barrier of geographical distance was erased. This resulted in always being up to date on what was going on in everybody else’s life, which by some was described in an ambivalent way. “Samuel”, in 9th grade, described being part of a chat group:

Samuel: *Well, it’ll be a little like being able … eh … you don’t need to meet to get it so … time with friends or how should I say it … if I’m going to meet my relatives then it may feel it would be more fun meeting friends then …but at the same time you’ve already spoken about most things before seeing them like … if there’s someone you haven’t met for ages then there’s loads to talk about but it won’t really be the same because you know what everybody’s done all the time … eh …*

Even though hanging out was mostly described in relation to friends, examples involving parents or other adults were given. When parents were mentioned, adolescents expressed cherishing activities alone with one parent, such as going for a run or preparing a meal together. What seemed equally important to getting some alone time with one parent was family time. Adolescents expressed how every family member needed to participate for it to be really appreciated and the activity was not that important, indicating that this increased connectedness in the family. Youth workers were brought up when adolescents described after school activities with friends. Hanging out with youth workers at the local leisure center was important because they socialized with everybody who was there, which contributed to a sense of connectedness between everybody present, despite just being acquaintances and of different ages.

#### 3.2.2. Having Things in Common

Sharing interests or hobbies with other people both led to expanding networks and acted as glue in adolescents’ relations and networks. The older adolescents described wider networks than the younger ones, for example, how they made new friends by having a summer job, which made them feel more socially confident. It was easier to engage in conversation about shared interests, which subsequently facilitated getting to know each other. The shared interest(s) then seemed to function as a self-sustaining source of connectedness. Adolescents had this experience with peers, but also with teachers, sports coaches, or other adults that they described as important. One girl described how shared interests with friends worked as glue even when she had left their network and original source of friendship, a local football team. “Jenny”, in 9th grade, stated: 

Jenny: *I think it’s a lot to do with having had others … it wasn’t just football in the relationship there were other things … that we could have fun together without it being about football … football wasn’t the most important …*

Feeling connected to others via sports or hobbies also made adolescents less nervous about making new friends within that context but having things in common was not limited to interests. When asked about what made best friends so special, some also mentioned personal characteristics, such as sharing the same sense of humor or values. One adolescent expressed how her religious beliefs made her feel connected to people through their shared faith but also disconnected from many people who were not religious. Moreover, some older adolescents mentioned how being politically active could be both a source of connectedness to like-minded people and an impediment to interacting with people on the other side of the political spectrum. 

Sharing a history together also increased connectedness. Adolescents spoke of how being childhood friends made them feel connected, even if they were not close friends anymore. For adolescents that had moved from their childhood home to another city, keeping in touch with older friends was important despite talking very rarely. Moreover, sharing a history with others could work as glue, such as keeping old friends close even though adolescents were aware of the bad influence they inflicted. One boy that had a troubled past described how two of his friends from that time still got into trouble from time to time but that he cherished their relationship despite the opinion of others. The family was a strong source of connectedness for obvious reasons and was described in different ways. Adolescents appreciated sharing an interest with a parent since it became a natural source for spending quality time, such as watching a documentary or a football game together or going for a run. 

Between siblings, having things in common could mean sharing experiences with parents, such as feeling that parents put the same demands on all siblings. This increased connectedness between all siblings since no one felt disfavored. Adolescents also seemed to be encouraged to take on hobbies that their siblings were doing since having things in common increased connectedness. Still, adolescents described how being a family itself entailed connectedness, regardless of having mutual interests or not. The same reasoning applied to the extended family. Adolescents expressed how they felt connected to relatives despite not sharing any interests or hobbies. To spend time with cousins, aunts, uncles, and grandparents meant being surrounded by people that loved you and recurrent gatherings during holidays were considered to strengthen that bond.

### 3.3. Maintaining Control

Maintaining control crystallized from adolescents reasoning around their own behavior, habits, decision making and motives, which made it clear that they were actively and subconsciously trying to maintain control in their social relations and networks. A loss of control was described to create negative feelings such as stress or distrust towards others. This category consisted of three sub-categories, “deciding for yourself”, “dealing with change” and “having social skills”.

#### 3.3.1. Deciding for Yourself

Being able to decide for yourself and having the freedom to do so was important for adolescents, but to maintain control it seemed more important to learn and be aware of the social boundaries that existed in different networks, and to act within these. Adolescents were aware of how power dynamics and their degree of freedom in networks affected the possibility to decide for themselves. For example, adolescents knew where their parents drew the line when it came to bringing home friends after school, staying out late, or when to do homework, and it was not described in a problematic way. It was when their degree of freedom was unclear or their ability to make sensible decisions was questioned that they reacted negatively. For example, having an overprotective sibling could mean that information was passed on to parents without consent, which affected the sense of freedom, resulting in a loss of control. Uncertainty could also arise in a friendship-like relationship with parents. “Josefin”, in 9th grade, who had divorced parents, described the relationship with her mother:

Josefin: *Well … sometimes she interferes a bit too much … and then it becomes as if … she is more a friend than she is a mother … bit it’s usually fairly even ….*

Spending time with grandparents was appreciated. Adolescents felt they had more freedom and were asked to decide on the activity to a greater extent than in other networks. This meant taking a break from some of the social boundaries that existed in other networks and a chance to act more like an adult. Adolescents found it encouraging to be given the freedom to show that they could make sensible decisions. Being allowed to stay out late with friends, going by bus to the city center or taking the train to visit relatives could be an important boost to confidence, but having the freedom to do those things also meant having the choice to refrain.

Adolescents talked about decision making in relation to the online setting as well. When interacting with friends, the opportunity to choose between group and private conversations brought a stronger sense of control. Whether or not to engage in conversations with strangers online was also described in an unproblematic way. The option to block, mute, or simply log off gave adolescents a feeling of power, which made them less cautious about who they interacted with.

#### 3.3.2. Dealing with Change

Adolescents expressed how changes in their life impacted their social relations and networks. Sometimes the change was easy to attach to a specific event, such as moving to a different city, changing schools, or parents splitting up after a divorce. Other changes were described more as processes that happened during a long period of time where adolescents were more involved in the process. Such changes included outgrowing friends or friends preferring the company of others or not enjoying the same activities as before. Older siblings moving out of the house was also described as a process where adolescents were aware of, and a part of, social changes in the family during a longer period of time. 

Adolescents had different ways of dealing with change. Some described a duality in relation to changes. While missing their old neighborhood and friends, they saw the benefits of moving to a different city and making new friends. Others felt sad about their parents getting a divorce but had adapted to switching between homes or were happy that their parents had met someone new. By reflecting on change and coming to terms with the new situation adolescents seemed to feel more in control. Some accepted changes in relationships and networks as something that naturally occurred, as experiences that they needed to manage and learn from. Others described how changes resulted in a loss of control, which triggered insecurity about the future. Being uncertain whether a network of friends was going to stick together in the foreseeable future was troublesome for some, whereas others accepted the fact that friends would come and go as they transitioned into adulthood. “Denise”, in 5th grade, stated:

Denise: *It can be, even though you’ve known one another a long time that you fall apart so … that you’re not the best of friends forever even if you’ve been so for five years …*

#### 3.3.3. Having Social Skills

Adolescents described conscious choices and actions that they used frequently in social relations and networks. These skills were learned through previous experiences and the incentive to use them seemed to be to steer outcomes or consequences in an anticipated direction, thus maintaining control. For example, adolescents were aware of differences between family members or friends in how they would respond to questions or what advice they would give on certain issues. This meant adolescents would choose carefully which family member or friend to address. 

Another social skill that adolescents used was to take on different roles or adapt their behavior depending on who they interacted with. Mostly this was described in relation to friends or more specifically when moving between different friend networks. However, it was not something that they felt compelled to do or that they associated with negative feelings but rather thought of as a helpful skill. Using their social skills could also mean to refrain from actions that would lead to a loss of control, such as ignoring annoying peers to prevent a dispute from escalating or letting a friend have their way in a discussion. Adolescents could choose not to engage if a peer was feeling miserable since they felt uncertain about their ability to help and instead involved an adult, such as a teacher, counsellor, or a parent. 

Apart from the conscious choices and actions, adolescents described something else that seemed to derive from a universal value or norm of wanting what was best for others. Even though an action in this sense could go against adolescents’ own interests or what was best for them at the time, they still chose to go through with it. Using these skills did not necessarily give immediate or long-term benefits but they increased adolescents’ sense of control. “Adnan”, in 9th grade, whose parents were born abroad, described how he did not want to worry his parents: 

Adnan: *Because I don’t want to say I feel bad in front of my mum or dad … because I don’t want them to be worried about … about me feeling good or bad … I just want to show that I’m happy and feel good …*

MA: *What would happen if they became worried then?*

Adnan: *Yes, but then … then they’d be so very over-protective and yes … I don’t actually know because I haven’t tried … so that’s interesting …*

Helping around the house and contributing to a positive climate in the family was important for adolescents, which sometimes meant putting the family before themselves. This included a desire that others, such as younger siblings, should be better off than themselves. Some adolescents described how being supportive, kind, and helpful was something they strived for in all their relations and networks. However, in reality this seemed to be easier said than done.

## 4. Discussion

This study aimed to explore social capital from the perspective of adolescents in relation to their mental health. Taken together, the findings revealed three key elements that were important to adolescents: access to a safe space, feeling connected to others, and maintaining control in their social interactions. Access to a safe space was identified as crucial, involving trust, honesty, and unconditional access in their closest relationships. Gender and age differences were observed in this aspect. Feeling connected to others played a significant role in shaping adolescents’ identity, fostering a sense of belonging and purpose. This extended to both close relations and broader contexts. Adolescents also emphasized the importance of maintaining control in their social interactions, actively and subconsciously seeking to control their behavior, decision making and motives. Loss of control was associated with negative emotions such as stress and distrust. In summary, these findings shed light on the role of social capital in adolescents’ mental health and provide insights for interventions and support systems aiming to enhance their well-being. 

This study portrays social capital in relation to mental health from the perspective of adolescents. This perspective is a fundamental addition to help fill in the gaps in interpretation of the increasing amount of quantitative research on social capital and adolescent mental health. The voices of the adolescents in this study provides a rich image that connects cognitive and structural aspects of the universal resources that is social capital for adolescents, i.e., trust, sense of belonging, reciprocity, and support [10], while also contributing with unique experiences, manifestations, and contextual relevance specific for adolescents in a modern contemporary welfare state.

The experiences of a safe space shared here relate well to what is commonly labeled bonding social capital, since it was attributed foremost to relationships within the family, the extended family, and close friends. The family is described in the literature as the primary source of bonding social capital [28,29]. It is, however, important to clearly delimit what constitutes the “family” and a definition should consider the sociocultural meaning of family [7]. The adolescents in this study referred primarily to parents and siblings as immediate family, but sometimes included stepfamily, grandparents, aunts, or cousins in that term. It may, thus, be appropriate to consider the wider definition of the term for Swedish adolescents. This notion of the term “family” extends previous qualitative research that showed how the wider family can be rich in bonding social capital and may offer support and guidance in difficult times [17]. Some of the adolescents also described how close relationships with non-parent adults offered a safe space. Sports coaches, teachers, and school counsellors were examples of adults that adolescents could confide in, especially if the family situation was troublesome. Other research supports the importance of adult role models as sources of guidance, support, and encouragement in adolescence, in particular for adolescents who experience adversity in other social contexts [30]. The safe space in peer networks revolved mostly around being able to share everything with like-minded people, with reference to age, gender, and values. This adds to what has been described previously about homogeneity in bonding peer networks among adolescents [31,32], but also touches on the downsides of social capital for adolescents, such as social exclusion [33] and behavioral contagion [6]. The dynamic nature was evident as the adolescents evaluated their safe spaces continuously. This pinpoints an important part of adolescence where social networks are more fluent and expand more rapidly than in other stages of life. The safe space, thus, offered a sense of safety and comfort in everyday life in its latent form, and was evaluated or considered whenever the adolescents faced adversity.

The category of feeling connected to others described relationships and networks of both bonding and bridging ties in multiple contexts. In bridging networks, an emphasis was placed on shared interests, social behavior, and the type of activity in order to benefit from interaction. Once a bonding tie was formed, values and expectations were more implicit, and the type of activity seemed less important. There was a latent form of connectedness that increased whenever interaction occurred. The adolescents described how hanging out with peers, regardless of activity, meant a great deal to them. Hanging out has been described in other research as an important part of life in adolescence that is facilitated by the limitlessness of the online arena [32]. In line with this, the online context was addressed foremost in this category, where adolescents described computer games and chat groups as ways to stay connected after school hours or during holidays. Doing activities on a regular basis, whether it was together with the family, peers or organized sports, the quantity seemed to matter, as well as being part of multiple overlapping networks. This was connected to a fear of loneliness and social exclusion, which has previously been described by Morrow [34]. Research has shown that the social environment in the family is related not only to depressive symptoms among young adolescents, but also to fear of missing out (FoMO), which triggers excessive internet use for example [35]. This category thus visualized structural components of social capital, aside from the evident cognitive components of sense of belonging and connectedness. 

The category maintaining control was about personal agency and navigating social relationships. The desire for control was strong, and its manifestations ranged from daily evaluation of relationships to avoidance of seeking support from parents because of uncertainty about their reaction and to spare them of emotional burden. This offered a rich image of the personal agency of adolescents today and their reasoning around their own behavior in relation to others. This can be connected to the dimension of social networks and sociability, as described by Schaefer-Mcdaniel [36] in her theoretically based conceptualization of social capital for young people. The notion of sociability, drawing on Bourdieu’s notion, is described as skills that help adolescents navigate social relationships to get ahead [36]. Portes [37] described social control from the external perspective, by suggesting that institutional and social rules and norms, upheld by social capital, give people a sense of control [37]. 

The findings of our study connect sociability and social control and add that social control is a product of past and present social interaction and the presence of norms and structures, and therefore, it goes beyond an ability. The strive to maintain control relied upon events that occurred not only within the immediate surroundings but also in adolescents’ extended networks, therefore reliant on both internal and external factors. Since social capital consists of resources, the resource proposed here to connect the category maintaining control to the concept of social capital is predictability. Predictability, as a resource in adolescents’ social relationships and networks, functions as a compass that seems to facilitate control. When events and their consequences become unpredictable, which occurs naturally with the expansion of networks in adolescence, many adolescents may not have the capacity or experience to navigate. Instead, having the possibility of returning to a predictable environment where actions and their consequences can be anticipated seems vital for adolescents in times of mental imbalance. Agency and the ability to act according to universal values constitute internal factors, which means that mental health problems would reduce predictability. External factors entail the mental health of others, the presence and nature of norms and social expectations and the occurrence of events that may disrupt existing structures and resources, such as parental divorce or a friend going behind your back.

To summarize, the latent constructs or resources of social capital, i.e., trust, sense of belong, support, etc., may be universal, but the socio-culturally specific manifestations can be better understood by involvement of the target group in qualitative research [16]. Qualitative methods may thus be beneficial in order conceptualize and operationalize social capital for young adolescents in a way that covers the breadth of social capital among adolescents. The social environment and the ways in which adolescents socialize and communicate have certainly changed over the past decades together with the societal innovations in mobility and communications and electronics, and therefore also the relationship with adolescent health and wellbeing [38]. The resource of predictability in social relationships and networks is here suggested to fill a gap that has emerged over the last thirty years. Adolescents today are exposed to a vastly increased amount of social information on a daily basis, while at the same time, they are expected to function as competent decision makers [39]. Nevertheless, they are more sensitive to social influence and relative comparison than other age groups [39,40], which arguably results in uncertainty and a strain on their mental health [41,42]. 

### Methodological Considerations

Trustworthiness in qualitative research should be reflected on in relation to the quality criteria of credibility, dependability, confirmability, and transferability [43,44,45]. Credibility refers to confidence in the findings and whether the findings represent a correct interpretation of the participants’ perspectives [43,44,45]. The credibility was strengthened in this study by a purposive school sampling technique of adolescents in grades 5 and 9. In all, 23 adolescents were interviewed, and the quality of the data was considered to be rich. Credibility was also strengthened by the authors (PS, JN) familiarity with the qualitative content design and the detailed descriptions provided of the data collection and analysis. The coding and analysis process was documented continuously to provide an overview and timeline of interpretations. One of the researchers (MA) had previously been engaged in an improvement project within these schools and was thus familiar with the context. Meanwhile, it is important to be aware of how one’s own attitudes, values, and biases may influence the research process. To strengthen credibility, joint discussions throughout the research process were held to enhance reflexivity, and all researchers were involved in and worked together during the analysis. Dependability refers to the stability of data and conditions [43,44,45]. Dependability was strengthened by the fact that the same researcher (MA) interviewed all adolescents, that the interviews started with the same main questions, and that the analysis process was in line with the standards for the qualitative content design. Confirmability concerns the aspect of neutrality and refers to the data representing the information provided by the participants and accurately reflects their voices [43,44,45]. In our study, confirmability was strengthened by the fact that the findings are grounded in the narratives of adolescents with diverse experiences and demonstrated by quotations that enhance and illuminate the content and offer the reader an opportunity to determine the trustworthiness of the data. Transferability refers to whether the findings can be transferred to other contexts or settings [43,44,45]. Although the sample in this study can be described as diverse, it is still within what may be considered “generic Swedish adolescents” that attend school, with no diagnosed mental illness and not living in a vulnerable or alienated community. The study was conducted at one single municipality in Sweden, and three schools were included, representing only townships and towns, which may be seen as a weakness. The schools were, however, chosen to represent a variety of socioeconomic levels. We have made efforts to provide a clear description of both the participants and the setting to enhance transparency and the evaluation of transferability. Further research may focus on adolescents living in larger cities or particular vulnerable groups of adolescents.

## 5. Conclusions

This study illustrates social capital in relation to mental health as described by adolescents in Sweden. Having access to a safe space is vital for adolescents’ mental health, by holding resources such as mutual trust, honesty, and unconditional access. Adolescents evaluate these relationships continuously, and it is evident that safes spaces are not static, showing the need for these types of bonds in multiple relationships. Feeling connected to others is important in close relationships and reveals the glue that holds networks together, but also portrays sociability, both being aspects that help shape the identity of adolescents and give them a sense of belonging. Maintaining control is about the ability and possibility to anticipate outcomes of actions, and to adapt to different networks and social contexts. Predictability is, thus, an important resource for mental health, influenced by both internal and external factors, which makes it an interesting subject for intervention and future research. Whether this is by increasing health literacy, sharing experiences, or following through on promises made, nurturing predictability in social networks may be an important resource in relation to the mental health of adolescents. The findings presented here may inform parents, practitioners, and adolescents themselves for building healthy relationships and promoting healthy social behavior. Future qualitative studies are needed to further elucidate sociocultural variation in the formation, utilization, and manifestation of social capital among young adolescents. Qualitative research may also help advance social capital theory and operationalization of social capital for adolescents in quantitative research. Conducting research with as opposed to about adolescents is encouraged from an ethical and quality perspective.

## Figures and Tables

**Table 1 ijerph-20-06223-t001:** Example of analysis showing condensed meaning units, codes, subcategories, and categories.

Condensed Meaning Unit	Code	Subcategory	Category
It feels good having siblings, you’ve always got someone you can go to.	Siblings are always there	Having someone close to you	Accessing a safe space
You’ve got something to do, you’re not alone and then it’s more fun, having friends is fun. I can always get in touch with them.	Can always get in touch with friends		
They usually play table tennis together, but I usually don’t join in. I could join in if I wanted to.	Being able to join in if one wants to	Being part of an inclusive and honest environment	
I can always say what I think with my friends.	Able to say what you think to your friends		

## Data Availability

The qualitative datasets generated and analyzed during the current study are not available.

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
