# Peer review of "Social Capital in Relation to Mental Health—The Voices of Adolescents in Sweden"

_ijerph, 2023, doi:10.3390/ijerph20136223_

Round 1

Reviewer 1 Report

Thank you for sending me this paper to review. I found it interesting and thoughtful and learned a great deal from reading it. I particularly liked the quality of the description and discussion of the findings. The topic is important and the observations made by the authors are helpful. I have very little to suggest that will improve the paper but make an observation that the authors might like to consider.

The presentation of results is very long and detailed. Whilst admirable in one sense, it also makes the argument sometimes difficult to follow. The key points that are being made have a tendency to get lost in the fullness and complexity of the interpretation. I wonder if this could be addressed by identifying the key points that authors want to make and shortening the discussion around those points. I know this is a difficult thing to do but think it would increase the chances of readers retaining they essential findings of this paper.  One example of this is that the conclusion to the paper suggests that predictability stands out as important in maintaining young people's mental health. I had not picked that up from my reading. On a related note, the authors suggest that the significance of predictability in social relations of adolescence is something that might be important to consider when developing interventions. An example of what this might look like would be helpful.

Though nicely written for the most part, paper would benefit from being read through carefully by someone for whom English is a first language. There are non-standard uses of English and small grammatical errors scattered throughout.

Author Response

Reviewer 1:

Thank you for sending me this paper to review. I found it interesting and thoughtful and learned a great deal from reading it. I particularly liked the quality of the description and discussion of the findings. The topic is important and the observations made by the authors are helpful. I have very little to suggest that will improve the paper but make an observation that the authors might like to consider.

Response: We appreciate the interest in the topic and paper.

The presentation of results is very long and detailed. Whilst admirable in one sense, it also makes the argument sometimes difficult to follow. The key points that are being made have a tendency to get lost in the fullness and complexity of the interpretation. I wonder if this could be addressed by identifying the key points that authors want to make and shortening the discussion around those points. I know this is a difficult thing to do but think it would increase the chances of readers retaining they essential findings of this paper.  One example of this is that the conclusion to the paper suggests that predictability stands out as important in maintaining young people's mental health. I had not picked that up from my reading. On a related note, the authors suggest that the significance of predictability in social relations of adolescence is something that might be important to consider when developing interventions. An example of what this might look like would be helpful.

Response: Thank you for bringing this to our attention. We have now introduced the discussion with a brief summary of the results highlighting the key points that are then further discussed. (1st paragraph of the discussion). In the discussion section, when the category “maintaining control” is discussed, we attempt to clarify the connection with social capital (sum of resources) by presenting “predictability” as the resource that social control relies on. See changes highlighted in yellow starting on line 585 and 663.

Though nicely written for the most part, paper would benefit from being read through carefully by someone for whom English is a first language. There are non-standard uses of English and small grammatical errors scattered throughout.

Response: The manuscript has undergone an additional review of grammatical errors and English language. We hope that this is to your satisfaction.

Reviewer 2 Report

This work addresses an issue that currently arouses great social interest: mental health in adolescence, on which we know that the pandemic has had a great impact. The perspective adopted around the concept of social capital seems adequate and is useful for the development of social and educational policies with an impact on improving mental health. In justifying the need for a qualitative approach, the authors use three reasons: the predominance of quantitative studies and the scarcity of qualitative ones, the time elapsed since the formulation of the construct by Bordieu in the 20th century and the changes in the social world of adolescents since then, and the lack of differentiation in many quantitative studies between the social capital of adults and that of adolescents. However, they do not offer us a true state of the art in relation to these three issues. Thus, it would be convenient for the authors to explain what the contributions of recent quantitative studies have been in relation to the social capital of adolescents and what limitations these methods offer in this regard, what changes have occurred in the context that influence the social capital of adolescents in recent decades, and what aspects of studies on social capital neglect the specificities of adolescence. This is important for the discussion of the results. But it is also relevant for the methodology that is going to be used, since it must be clarified if the assumptions generated by the theory are based on, or if the inquiry is approached with a strategy typical of Grounded Theory.

As regards the methodology, the steps followed in the information collection and especially analysis processes seem well-founded and well explained.. But there is an aspect on which the authors go unsatisfactorily: the characterization of the sample. Both in exposing the method and in the discussion, the authors state “The adolescents represented both boys and girls, different ages, levels of socioeconomic status, family constellations, religious views and different living conditions”. This characterization is tremendously generic, and reading the results seems to call into question the diversity of the students. Being an investigation on social capital, the lack of information on the social context of the schools, on the existence or not of cultural diversity and origins of the students is not explained. Many of the adolescent expressions used to illustrate the research correspond to individualistic values, fully established in the majority of autochthonous European families. But in many of those who come from other continents, or are part of minority groups, such as the Roma, educational goals are oriented more towards interdependence than absolute autonomy, and some research has shown that this is clearly reflected in the values of adolescents or in their conceptualization of the notion of independence. Neither do direct references to situations of social exclusion or risk of dropping out of school appear in the results, threats that remain over sectors present in all European societies. In both cases (belonging to non-hegemonic cultures or risk of social exclusion) they contribute to very different configurations of adolescents' social capital. Nor does there seem to be a presence of adolescents with disabilities, or with school failure problems, or victims of bullying, or victims of family abuse or with addictive behaviors. Neither does it seem to pay attention to the diversities of sexual option and gender identity. All this does not nullify the interest of the investigation, but compromises any reading that seeks to generalize the results. Reading these seems to reveal a placid social environment, without social barriers, with well-structured families and integrated into the social system, and in fact it seems that this is a condition for the type of social capital built by the subjects. Not noticing this can lead to erroneous generalizations or, what is worse, contribute to the definition of a “normative” adolescence that throws other realities into the field of the pathological in relation to the construction of social capital.

Thus, it should be noted if the subjects are framed in the hegemonic culture, if they are all children of native families, if they do not suffer from the risk of exclusion or early school leaving, if they do not suffer from any disability, if no relevant elements have been found. regarding sexual choices or gender identity…

The only reference to the fact that precautions must be maintained in the face of the "placid panorama" that the interviews seem to reflect, in which there are no hints of conflict (precisely something that had been at the heart of the definitions of adolescence until the eighties of the past century) seems to be just the final sentence of the article: “However, in reality this seemed to be easier said than done”.

In summary, I recommend the publication of the article after some changes, mainly, completing the introduction to justify in which specific aspects it will contribute to the theoretical knowledge about the social capital of adolescents and its contribution to mental health. Above all, clarifying the social context of the adolescents interviewed, stating how the results respond to the characteristics of that cultural, social and personal context of themselves. And explaining the limitations of the study in relation to the diversity of evolutionary transits through adolescence.

Author Response

Reviewer 2:

This work addresses an issue that currently arouses great social interest: mental health in adolescence, on which we know that the pandemic has had a great impact. The perspective adopted around the concept of social capital seems adequate and is useful for the development of social and educational policies with an impact on improving mental health.

Response: We appreciate the interest in the topic and paper.

In justifying the need for a qualitative approach, the authors use three reasons: the predominance of quantitative studies and the scarcity of qualitative ones, the time elapsed since the formulation of the construct by Bordieu in the 20th century and the changes in the social world of adolescents since then, and the lack of differentiation in many quantitative studies between the social capital of adults and that of adolescents. However, they do not offer us a true state of the art in relation to these three issues. Thus, it would be convenient for the authors to explain what the contributions of recent quantitative studies have been in relation to the social capital of adolescents and what limitations these methods offer in this regard, what changes have occurred in the context that influence the social capital of adolescents in recent decades, and what aspects of studies on social capital neglect the specificities of adolescence. This is important for the discussion of the results.

Response: Thank you for your comment. We have in the background section added state of the art in relation to the three issues that you highlighted. Thus, we have described what the contributions of recent quantitative studies have been in relation to the social capital of adolescents and what limitations these methods offer in this regard, see text highlighted in yellow starting on line 41. We have also tried to clarify the changes that have occurred in the context and how this has influenced social interaction in the past two decades, and what previous research has found to be problematic with the lack of distinction between adult and adolescent social capital. See line 61.

But it is also relevant for the methodology that is going to be used, since it must be clarified if the assumptions generated by the theory are based on, or if the inquiry is approached with a strategy typical of Grounded Theory.

Response: We fully understand why this question may have arisen; however, our intention was never to create a new theory of social capital. Instead, the study wanted to increase the knowledge of sociocultural specific manifestations of social capital and to understand adolescents’ perspectives on what resources they find to be important in relation to their mental health. We intended to be close to the stories that the adolescents shared and therefore chose a qualitative content analysis.  

As regards the methodology, the steps followed in the information collection and especially analysis processes seem well-founded and well explained. But there is an aspect on which the authors go unsatisfactorily: the characterization of the sample. Both in exposing the method and in the discussion, the authors state “The adolescents represented both boys and girls, different ages, levels of socioeconomic status, family constellations, religious views and different living conditions”. This characterization is tremendously generic, and reading the results seems to call into question the diversity of the students. Being an investigation on social capital, the lack of information on the social context of the schools, on the existence or not of cultural diversity and origins of the students is not explained. Many of the adolescent expressions used to illustrate the research correspond to individualistic values, fully established in the majority of autochthonous European families. But in many of those who come from other continents, or are part of minority groups, such as the Roma, educational goals are oriented more towards interdependence than absolute autonomy, and some research has shown that this is clearly reflected in the values of adolescents or in their conceptualization of the notion of independence. Neither do direct references to situations of social exclusion or risk of dropping out of school appear in the results, threats that remain over sectors present in all European societies. In both cases (belonging to non-hegemonic cultures or risk of social exclusion) they contribute to very different configurations of adolescents' social capital. Nor does there seem to be a presence of adolescents with disabilities, or with school failure problems, or victims of bullying, or victims of family abuse or with addictive behaviors. Neither does it seem to pay attention to the diversities of sexual option and gender identity. All this does not nullify the interest of the investigation, but compromises any reading that seeks to generalize the results. Reading these seems to reveal a placid social environment, without social barriers, with well-structured families and integrated into the social system, and in fact it seems that this is a condition for the type of social capital built by the subjects. Not noticing this can lead to erroneous generalizations or, what is worse, contribute to the definition of a “normative” adolescence that throws other realities into the field of the pathological in relation to the construction of social capital.Thus, it should be noted if the subjects are framed in the hegemonic culture, if they are all children of native families, if they do not suffer from the risk of exclusion or early school leaving, if they do not suffer from any disability, if no relevant elements have been found. regarding sexual choices or gender identity…

Response: Thank you for this reflection. We have now updated the method section to provide a clearer view of the setting and about the characteristics of the sample and tried to clarify the social context of the adolescents in the results section. We have also elaborated in the methodological considerations in relation to transferability of the results. See text highlighted in yellow starting on line 119 and 164.

The only reference to the fact that precautions must be maintained in the face of the "placid panorama" that the interviews seem to reflect, in which there are no hints of conflict (precisely something that had been at the heart of the definitions of adolescence until the eighties of the past century) seems to be just the final sentence of the article: “However, in reality this seemed to be easier said than done”.

Response: We understand this reflection and have attempted to provide more context to the citations in the results section. We are a bit uncertain if there is anything else we can do to improve the manuscript based on this comment. It may be so that the original research question, the design, aim and sampling technique in this study are more indicative of an asset approach rather than a deficit approach, by focusing on relationships that the adolescents felt was important for their mental health and having them describe the resources there, as oppose to the opposite. Still, we feel that the results hold important descriptions of the struggles that adolescents face. See text highlighted in yellow in the results section where we have provided a bit more context to the citations.

Response:

In summary, I recommend the publication of the article after some changes, mainly, completing the introduction to justify in which specific aspects it will contribute to the theoretical knowledge about the social capital of adolescents and its contribution to mental health. Above all, clarifying the social context of the adolescents interviewed, stating how the results respond to the characteristics of that cultural, social and personal context of themselves. And explaining the limitations of the study in relation to the diversity of evolutionary transits through adolescence.

Response: Thank you, we have addressed these suggestions in the manuscript. For detailed answer to each question, please see the responses above.

Reviewer 3 Report

The manuscript presents an important research topic of interest, particularly listening to the voice of adolescents within the context of their social capital. As such, the manuscript makes an important contribution.

The qualitative study is thoughtfully designed and the methodology is satisfactory. It is a well written manuscript.

I offer some minor points for the authors' consideration -

1. That the demographic data is presented in a Table rather than a narrative paragraph for ease of review and reading, e.g. age groups, gender, school year, socio-economic status and family profile, etc.

2. It would be instructive to have some commentary on whether socio-economic status etc have any influence on the results of the study.

3.  I found section 4.1 Methodological Considerations under the Discussion section to be awkwardly placed. On reading the section, I suggest that 'Methodological Considerations' is better placed within the methodology section of the manuscript. In the Discussion, it would be useful to appreciate whether the Swedish study findings are similar to other studies.

4. Finally, it would be instructive to elaborate on the Conclusion in terms of practice and future research considerations.

Thank you for the opportunity to review the manuscript.

No further comments on the quality of the English language. My suggestion would be to break up the paragraphs as most of the paragraphs are rather long with two or more themes.

Author Response

Reviewer 3:

The manuscript presents an important research topic of interest, particularly listening to the voice of adolescents within the context of their social capital. As such, the manuscript makes an important contribution.The qualitative study is thoughtfully designed and the methodology is satisfactory. It is a well written manuscript.

Response: Thank you for your kind words. We appreciate the interest in the topic and paper.

I offer some minor points for the authors' consideration -

  1. That the demographic data is presented in a Table rather than a narrative paragraph for ease of review and reading, e.g. age groups, gender, school year, socio-economic status and family profile, etc.

Response: We are afraid that this is not possible except for age and gender since demographic data were not gathered systematically beforehand but from each individual interview and of different quality depending on what the adolescent felt comfortable sharing in their initial description of themselves and during the interview. This, however, may be considered a limitation, and we have therefore added a couple of sentences about this in the methodological consideration. See text highlighted in yellow starting on line 724.

  1. It would be instructive to have some commentary on whether socio-economic status etc have any influence on the results of the study.

Response: Thanks for comment. We have added more information about the setting and characteristics of the sample and tried to clarify the social context of the adolescents interviewed, see line 118 and 164. However, based on the design of the study and a relatively small sample size we could not say anything about whether socio-economic status etc had any influence on the results of the study. We can only conclude that socio-economic status varied between participants based on the interviews and thus each reader should reflect on the transferability of the results with regard for different socioeconomic levels. See line 720 and forward under methodological considerations.

  1. I found section 4.1 Methodological Considerations under the Discussion section to be awkwardly placed. On reading the section, I suggest that 'Methodological Considerations' is better placed within the methodology section of the manuscript. In the Discussion, it would be useful to appreciate whether the Swedish study findings are similar to other studies.

Response: We understand your concern. Based on the journal’s guidelines, the methodological considerations are usually placed in the discussion section. Therefore, we have not made any changes. If the Editor would agree with your suggestion, we are happy to place the section differently in the article.

  1. Finally, it would be instructive to elaborate on the Conclusion in terms of practice and future research considerations.

Response: Thank you for your suggestions. We have developed the conclusion and added implications for practiced and future research. See text highlighted in yellow starting on 742.

No further comments on the quality of the English language. My suggestion would be to break up the paragraphs as most of the paragraphs are rather long with two or more themes

Response: Thank you, We have changed that in line with your suggestion.

Round 2

Reviewer 2 Report

Although I believe that the research does not adequately identify diversity in the adolescent population, I believe that it may be a study that adequately accounts for the mainstream population in your country, pending comparative studies that take this diversity into account.
The rest of the formal problems were fixed in the new version.